# Genome-Wide Identification and Comparative Transcriptome Methods Reveal *FaMDHAR50* Regulating Ascorbic Acid Regeneration and Quality Formation of Strawberry Fruits

**DOI:** 10.3390/ijms24119510

**Published:** 2023-05-30

**Authors:** Guoyan Hou, Min Yang, Caixia He, Yuyan Jiang, Yuting Peng, Musha She, Xin Li, Qing Chen, Mengyao Li, Yong Zhang, Yuanxiu Lin, Yunting Zhang, Yan Wang, Wen He, Xiaorong Wang, Haoru Tang, Ya Luo

**Affiliations:** College of Horticulture, Sichuan Agricultural University, Chengdu 611130, China; 2021205002@stu.sicau.edu.cn (G.H.); 2021101003@stu.sicau.edu.cn (M.Y.); 2022305087@stu.sicau.edu.cn (C.H.); jiangyuyan@stu.sicau.edu.cn (Y.J.); 2022205007@stu.sicau.edu.cn (Y.P.); 2022205078@stu.sicau.edu.cn (M.S.); 18684016131@163.com (X.L.); supnovel@sicau.edu.cn (Q.C.); limy@sicau.edu.cn (M.L.); zhyong@sicau.edu.cn (Y.Z.); linyx@sicau.edu.cn (Y.L.); asyunting@sicau.edu.cn (Y.Z.); wangyanwxy@sicau.edu.cn (Y.W.); hewen0724@gmail.com (W.H.); wangxr@sicau.edu.cn (X.W.); htang@sicau.edu.cn (H.T.)

**Keywords:** strawberry, ascorbic acid regeneration, *FaMDHAR50*, fruit quality, genome-wide analysis, sugar and acid metabolism, anthocyanin accumulation

## Abstract

Ascorbic acid (AsA) is a crucial water-soluble antioxidant in strawberry fruit, but limited research is currently available on the identification and functional validation of key genes involved in AsA metabolism in strawberries. This study analyzed the FaMDHAR gene family identification, which includes 168 genes. Most of the products of these genes are predicted to exist in the chloroplast and cytoplasm. The promoter region is rich in cis-acting elements related to plant growth and development, stress and light response. Meanwhile, the key gene *FaMDHAR50* that positively regulates AsA regeneration was identified through comparative transcriptome analysis of ‘Benihoppe’ strawberry (WT) and its natural mutant (MT) with high AsA content (83 mg/100 g FW). The transient overexpression experiment further showed that overexpression of *FaMDHAR50* significantly enhanced the AsA content by 38% in strawberry fruit, with the upregulated expression of structural genes involved in AsA biosynthesis (*FaGalUR* and *FaGalLDH*) and recycling and degradation (*FaAPX*, *FaAO* and *FaDHAR*) compared with that of the control. Moreover, increased sugar (sucrose, glucose and fructose) contents and decreased firmness and citric acid contents were observed in the overexpressed fruit, which were accompanied by the upregulation of *FaSNS*, *FaSPS*, *FaCEL1* and *FaACL*, as well as the downregulation of *FaCS*. Additionally, the content of pelargonidin 3-glucoside markedly decreased, while cyanidin chloride increased significantly. In summary, *FaMDHAR50* is a key positive regulatory gene involved in AsA regeneration in strawberry fruit, which also plays an important role in the formation of fruit flavor, apperance and texture during strawberry fruit ripening.

## 1. Introduction

Ascorbic acid (AsA), also known as vitamin C, is a powerful antioxidant found in plants [1]. AsA plays a vital role in plant growth, development [2], and stress response, protecting the plant from environmental stressors such as UV radiation, drought, and pathogens [3,4]. Additionally, AsA has numerous health benefits for humans, such as boosting the immune system, promoting wound healing, and preventing chronic diseases [5]. Due to the absence of the gene encoding l-guluronic acid-1,4-lactone in humans, the human body is unable to synthesize AsA [6]. Therefore, humans can only obtain AsA through supplementation from fruits and vegetables. Thus, increasing the content of AsA in horticultural products has always been a significant research focus.

The synthesis of AsA in plants involves multiple pathways, including the L-galactose pathway, the L-gulose pathway, the myoinositol pathway, and the d-galacturonate pathway. The L-galactose pathway is recognized as the major pathway for AsA synthesis in horticultural crops such as tomato, kiwifruit, apple, and jujube [7,8]. In the L-galactose pathway, D-glucose 6-P is first converted to D-mannose-1-P by enzymes such as phosphoglucose isomerase (PGI), phosphomannose isomerase (PMI), and phosphomannomutase (PMM). After a series of reactions, L-galactono-1,4-lactone is generated [9], which is then catalyzed by L-galactono-1,4-lactone dehydrogenase (GalLDH) to produce AsA within the mitochondrial membrane [10]. GDP-D-mannose can also be converted to GDP-L-gulose by GME, and GDP-L-gulose can undergo a series of reactions to ultimately form L-gulono-1,4-lactone, which is then catalyzed by L-gulono-1,4-lactone oxidase (GulLO) to produce AsA in the L-gulose pathway [11,12]. L-gulono-1,4-lactone is also the final precursor of AsA generated by the myoinositol pathway and is converted to AsA under the catalysis of GulLO [13]. The D-galacturonate pathway was first discovered in strawberries [14], where methyl-D-galacturonate can be converted to D-galacturonate, L-galactonate, L-galactono-1,4-lactone, and AsA by methylesterase, D-galacturonate reductase (GalUR), aldono-lactonase (Alase), and GalLDH, respectively.

In addition to the above-mentioned AsA synthesis pathway, the content of AsA is also highly regulated by the AsA-GSH cycle, which is a key pathway involved in the regeneration of oxidized AsA. The AsA-GSH cycle involves the oxidation and dehydrogenation of AsA, catalyzed by ascorbate peroxidase (APX) and ascorbate oxidase (AO), resulting in the formation of monodehydroascorbic acid (MDHA). MDHA is then converted to dehydroascorbic acid (DHA) through a non-enzymatic disproportionation reaction and then regenerated to AsA through the catalysis of monodehydroascorbate reductase (MDHAR) and dehydroascorbate reductase (DHAR) [15]. Positive regulation of DHAR in AsA regeneration has been reported in several studies, including those of *Arabidopsis* [16], kiwifruit [17], and tomato [18]. However, there is still some controversy about the role of MDHAR in regulating AsA accumulation in fruits. For example, overexpression of the *BcMDHAR* gene from cabbage in tobacco led to a decrease in AsA content [19], while overexpression of the *MDHAR* gene from cherry in tobacco resulted in higher levels of AsA content [20]. The functionally deficient mutant of the *MDHAR4* gene in *Arabidopsis thaliana* had a decreased ratio of ascorbic acid to dehydroascorbate but did not affect the content of total AsA [21]. This indicates that the *MDHAR* gene family members have a regulatory role in AsA metabolism, but the key genes that play a role may vary between species. In addition, the recycling and regeneration of AsA are also regulated by transcription factors. One of the members of the plant-specific Dof (DNA-binding with one finger) family, *Dof 22*, and the tomato resistance protein *SlNL33* have recently been identified as negative transcriptional regulators of AsA cycling in tomatoes [22,23]. The *PbrMYB5* gene in pear has been shown to positively regulate the transcription of genes involved in AsA recycling and stress response, including *MDHAR*, *DHAR2*, and *APX* [24].

Strawberry (*Fragaria × ananassa* Duch.) is a perennial herbaceous plant belonging to the Rosaceae family and is highly regarded as the “queen of fruits” due to its unique flavor, rich nutrients, and antioxidant content [25]. The content of AsA gradually accumulates as the fruit ripens. Although genes such as *FaGalUR* [26], *FaAKR23* [27], and *FaMDHAR* [28], have been reported to be involved in the synthesis and metabolism of AsA in strawberries, the regulatory mechanism of AsA accumulation in strawberry fruit is still very limited. ‘Benihoppe’ is currently the most widely cultivated strawberry variety in China due to its high yield and excellent overall quality. During previous experimental research on strawberries, a high AsA variation material (Mutant, MT) was discovered in ‘Benihoppe’ (Wild, WT). Throughout the entire fruit development stage, the AsA content of MT was significantly higher than that of WT. However, the specific reason for regulating the differential accumulation of AsA is not yet clear. In this study, genome-wide identification and a comparative transcriptome approach were used to identify the key gene responsible for AsA regeneration. The results showed that *FaMDHAR50* is a key gene involved in the regulation of AsA accumulation and quality formation of strawberry fruit. The findings of this study provide crucial genetic resources for developing high-Vc strawberry cultivars and shed new light on strawberry fruit quality control.

## 2. Results

### 2.1. Genome-Wide Identification and Characterization of FaMDHAR Gene

A total of 168 *FaMDHAR* genes were identified based on the whole strawberry genome (Figure 1), and they were named *MDHAR1* to *MDHAR168* based on their chromosomal localization and distribution position (Appendix A). Analysis of the physicochemical properties of FaMDHAR (Appendix A) showed that the amino acid composition of FaMDHAR varied from 93 to 1081, and the molecular weight ranged from 9903.43 to 121,777.26 Da. Additionally, the theoretical isoelectric points ranged from 4.61 (FaMDHADR64) to 9.61 (FaMDHAR98). The instability coefficients varied from 16.42 to 56.17, with 75 proteins having instability coefficients greater than 40, indicating that almost half of the proteins were in an unstable state, while the remaining 95 proteins were stable. The hydrophobicity prediction results showed that the GRAVY values ranged from −0.639 to 0.523. Among the 168 proteins, 150 had GRAVY values less than 0, and only 18 proteins had GRAVY values greater than 0, indicating that most FaMDHAR were hydrophilic proteins. Furthermore, the predicted subcellular localization results showed that some products of the MDHAR genes were localized in organelles such as mitochondria, nuclei, and vesicles, while most products were observed in chloroplasts and cytoplasm. These 168 genes were widely distributed on 28 chromosomes, with the largest number of genes on four chromosomes, namely Fvb6-1, Fvb6-2, Fvb6-3, and Fvb6-4 (Figure 1).

### 2.2. Phylogenetic Analysis of FaMDHAR

To investigate the evolutionary relationships among MDHAR proteins in the whole genome of strawberry, a phylogenetic tree was constructed using MDHAR protein sequences from strawberry (n = 168) (Figure 2). The results showed that FaMDHAR could be divided into three subfamilies, namely FaMDHAR-a, FaMDHAR-b, and FaMDHAR-c. Among these subfamilies, FaMDHAR-a had the most members with 79, accounting for 47% of the total. On the other hand, the FaMDHAR-b family had the fewest members, with only 39 of them.

### 2.3. Analysis of Cis-Acting Elements in the Promoter Region of the FaMDHAR Gene

To explore the potential biological functions of the *FaMDHAR* gene, an analysis of cis-acting elements in the promoter region was conducted. A total of 70 cis-acting elements were identified, with most related to plant growth and development, phytohormone response, and biotic and abiotic stress responses (Figure 3). Moreover, light response elements (G-box, Box 4, and I-box) were highly enriched in the promoter. Of these, Box 4, G-box, and I-box elements appeared 371, 154, and 68 times, respectively, indicating a potential role in light response mechanisms. Furthermore, hormone elements such as abscisic acid response element (ABRE) and growth hormone response element (TGA-Box) were also abundantly enriched. In total, 315 ABRE abscisic acid response elements were enriched in 116 *FaMDHAR* genes, suggesting that *MDHAR* genes may play a crucial role in multiple signaling pathways of plant hormones, especially in the ABA pathway. In addition, a total of 322 biotic and abiotic stress response elements (ARE), 171 MBS elements, and a small number of trauma and pathogen response elements (WUN-motif) were enriched. These findings suggested that *FaMDHAR* may also play a role in biotic and abiotic stress responses, as well as in the defense and repair processes associated with trauma and pathogen invasion.

### 2.4. Characteristics of Phenotype and Ascorbic Acid Accumulation in MT and ‘Benihoppe’

There were significant phenotypic differences in the appearance of flowers and fruits between ‘Benihoppe’ and its MT (Figure 4A). The flowers of MT were larger in size for both corolla and sepals, with serrations observed in the MT. In addition, the fruits of MT were consistently smaller during development and ripening compared to ‘Benihoppe’. However, MT fruits displayed a reddish-black color, while ‘Benihoppe’ exhibited a bright red color at the full-red stage. We also conducted measurements of ascorbic acid content during fruit development and discovered that it increased gradually. Moreover, the ascorbic acid content of MT was significantly higher than that of ‘Benihoppe’ at all stages of fruit development and ripening (Figure 4B). Although the difference in dehydroascorbic acid content was not significant, the total ascorbic acid content reached 834.93 μg/g at the full-red stage, which was 1.2 times higher than that of ‘Benihoppe’.

### 2.5. Transcriptional Differences between ‘Benihoppe’ and MT in the AsA Metabolic Pathway

To better understand the molecular mechanisms behind the significant differences in AsA content between ‘Benihoppe’ and MT, we conducted a transcriptome analysis of the AsA metabolic pathway in both materials during fruit ripening. A total of eight differentially expressed transcripts were detected in the entire AsA anabolic pathway, including *GalUR*, *PMM*, *GGP*, *AO*, *APX*, *MDHAR*, *DHAR*, and *IMP* (Figure 5). In terms of the L-galactose pathway, two differentially expressed transcripts (*PMM, GGP*) were detected, while only one (*GalUR, IMP*) was detected for both the D-galacturonate pathway and the myoinositol pathway. No differential transcripts were detected in the L-gulose pathway. In the D-galacturonate pathway synthesis pathway, GalUR plays a key role in converting D-galacturonate to L-galactonate, followed by Alase and GalLDH to form AsA. During the three stages of LG, PR, and FR fruit in MT, the expression of *GalUR* (Fxac_16g30570, Fxac_14g01870, Fxac_15g01550) was significantly higher than that in ‘Benihoppe’. After AsA is synthesized, it can be metabolized to MDHA by APX and AO, and MDHA can be further converted to AsA by MDHAR, or non-enzymatically dismutated to DHA. In this experiment, it was found that the expression level of *AO* (Fxac_6g21870) in MT fruit was significantly lower than that in ‘Benihoppe’ fruit at three developmental stages. Conversely, the expression level of each transcript of *APX* showed inconsistencies in different developmental stages. Specifically, at the LG stage, the expression levels of FxaC_12g12170 and FxaC_10g381200 transcripts in MT were significantly higher than those in ‘Benihoppe’. In the PR stage, the expression of FxaC_11g35420 and FxaC_12g12170 transcripts in MT was significantly lower than that of ‘Benihoppe’. In the FR stage, the expression of FxaC_9g15040, FxaC_12g12170, and FxaC_10g38120 was significantly lower than that of ‘Benihoppe’. The APX gene transcript (FxaC_9g15040) showed the highest expression level in the FR stage in MT. It is worth noting that multiple transcripts of the *MHDAR* were detected to be significantly higher in all three developmental stages of MT fruit. Among these transcripts, two (Fxac_6g42700 and Fxac_7g41490) showed a significant increase in expression level in MT, with a 200% and 100% increase, respectively, in the LG stage, a 275% and 46% increase in the PR stage, and a 233% and 145% increase in the FR stage when compared to ‘Benihoppe’. DHA can be converted to AsA through two consecutive dehydrogenation steps, with the help of DHAR catalysis. During the LG and FR stages, the expression level of DHAR (FxaC_4g02930) was significantly higher in MT compared to ‘Benihoppe’. However, a decrease in expression was observed during the PR stage. Taken together, the significant increase in expression of *MDHAR* genes (Fxac_6g42700, Fxac_7g41490) in the AsA regeneration cycle promotes the replenishment of MDHA, resulting in higher levels of AsA accumulation in MT fruits.

### 2.6. Transient Overexpression of FaMDHAR50 Significantly Increases AsA Content in Strawberry Fruit

In order to investigate the effect of the strawberry *MDHAR* gene on AsA metabolism and accumulation, we selected the *FaMDHAR50* gene (Fxac_6g42700) for further functional validation. Overexpression of *FaMDHAR50* significantly inhibited the coloring of strawberry fruit (Figure 6A). The qRT-PCR results showed that 35s::FaMDHAR50 was overexpressed 39 times compared to the empty controls (Figure 6B), indicating successful overexpression. Overexpression of *FaMDHAR50* significantly decreased the firmness of strawberry fruit and the *a** value of strawberry fruit exterior (Figure 6C,D). Notably, compared to the control, the *FaMDHAR50*-overexpressed fruit exhibited a remarkable 38% increase in AsA content, with no significant difference in dehydrogenated AsA (Figure 6E). Besides, overexpression of *FaMDHAR50* led to a noticeable increase in the content of pelargonidin 3-glucoside, while the content of cyanidin chloride was significantly reduced. Consequently, the overall anthocyanin content in *FaMDHAR50*-overexpressed fruit decreased by 10% (Figure 6D). Furthermore, the sugar content in *FaMDHAR50*-overexpressed fruit was significantly higher than that in control, with sucrose, fructose, and glucose increasing by 48%, 25%, and 35%, respectively (Figure 6G). Meanwhile, the citric acid content of *FaMDHAR50*-overexpressed fruit decreased significantly by 19% (Figure 6H). These results indicated that *FaMDHAR50* is involved in regulating the metabolism of AsA, sugar, acid, and anthocyanin in strawberry fruit.

### 2.7. Quantification of Fruit Quality-Related Genes

To investigate the possible molecular mechanisms of *FaMDHAR50* in regulating strawberry fruit quality formation, we performed real-time fluorescence quantitative analysis on genes related to AsA metabolism, citric acid synthesis metabolism, the anthocyanin synthesis pathway, and the ABA pathway(Figure 7). The results showed that overexpression of *FaMDHAR50* led to a substantial increase in the expression levels of key enzymes involved in the AsA metabolic pathway. Specifically, the expression levels of *GalLDH*, *GalUR*, *DHAR*, *APX*, and *AO* were increased by 2.9, 4.7, 2.8, 1.7, and 2.8 times, respectively, compared to the control. Meanwhile, the expression of endo-1,4-beta-glucanase (*CEL1*) involved in fruit softening was significantly increased by 1.6 times, indicating that it may be a contributor to the softened texture. Both sucrose synthase (*SNS*) and sucrose phosphate synthase (*SPS*) can positively promote the synthesis of sucrose. In this study, the expression levels of both *SNS* and *SPS* were significantly increased, which were 12 and 1.9 times higher than that of the control. Furthermore, the expression of citrate synthetase (*CS*) involved in citric acid synthesis decreased by 50%, while the expression of citrate lyase (*ACL*) involved in citric acid degradation increased by 30%, potentially leading to the reduced citric acid content in *FaMDHAR50* overexpressed fruit. In the anthocyanin synthesis pathway, the expression of flavonoid 3′,5′-hydroxylase (*F3′5′H*) was significantly increased, which was 2.5 times higher than that of the control. However, the expression of anthocyanidin synthase (*ANS*) and dihydroflavonol 4-reductase (*DFR*), which directly catalyze anthocyanins, decreased by 30% and 40%, respectively, compared to the control. Meanwhile, the expression of flavonoid 3′-hydroxylase (*F3′H*) and UDP-glycose flavonoid glycosyltransferase (*UFGT*) was not significantly different from that of the control. Additionally, the expressions of 9-cis-epoxycarotenoid dioxygenase 1 (*NCED1*), the key genes of ABA glycosyl groups, were significantly elevated by 3.7 times that of the control, respectively. The above results suggested that upregulation of *FaMDHAR50* has a positive role in the expression of genes involved in fruit softening, AsA, and sugar accumulation, as well as the reduction of citric acid and anthocyanin content.

## 3. Discussion

*MDAHR* is an important enzyme for AsA regeneration in plants. *MDHAR* gene family members have been identified in plants such as peas, tomatoes, spinach, cucumber, soybean, and potato [29,30,31,32]. In this study, we identified a total of 168 *MDHAR* genes in cultivated strawberry, which is a larger amount than those previously identified in cotton [33] and *Arabidopsis* [34]. This may be attributed to cultivated strawberries being octoploid and therefore having a higher number of alleles. Multiple sequence alignments and phylogenetic tree analysis showed that the members of the *FaMDHAR* family can be divided into three groups. The function of plant MDHAR within the same group may be consistent. Group A has significantly more members than the other two groups, which is similar to that in cotton [33]. The products of the MDHAR genes are widely distributed in different organelles such as chloroplasts, mitochondria, peroxisomes, and cytosol [35,36]. The 168 MDHARs identified in strawberry were also found to be widely distributed in various organelles, especially in the chloroplast and cytoplasm (Appendix A). In this study, a large number of cis-acting elements related to plant growth and development, phytohormones, stress, and light responses were found to be enriched in the *FaMDHAR* promoter region, suggesting the involvement of *MDHAR* genes in various biological activities such as biotic and abiotic stress responses, oxidative stress and growth and development regulation [37].

With consumers increasingly focusing on the functional components of horticultural products, researchers are actively promoting functional strawberry breeding. Strawberry fruit is rich in vitamin C, and the content of AsA varies among genotypes. Currently reported AsA content of cultivated strawberry varieties ranges from 50–110 mg/100 g FW [38]. As is well known, strawberry fruits with high levels of ascorbic acid have many beneficial effects on human health. In this study, it was found that the AsA content of strawberry fruit gradually accumulated with the development and ripening of the fruit, reaching its highest level at the full red stage (Figure 4B). Moreover, the AsA content of MT is up to 83 mg/100 g FW in fully ripe fruits, which is 1.2 times higher than that of ‘Benihoppe’ (Figure 4B). The relatively higher expression levels of *FaMDHAR50* and *FaMDHAR26* during strawberry fruit development indicated that both of them may play an important role in strawberry fruit development. Interestingly, the analysis of differential gene expression related to AsA metabolism showed that *FaMDHAR50* may be one of the key genes regulating the accumulation of AsA in the mutants. Transient overexpression of *FaMDHAR50* resulted in a significant increase in both AsA and total AsA contents by 38% and 33% in strawberry fruits, respectively, compared with the control (Figure 6E). Meanwhile, the expression levels of key structural genes (*FaGalLDH*, *FaGalUR*, and *FaMDHAR*) involved in the synthesis and recycling of AsA were significantly enhanced (Figure 7). Although there is no direct evidence that *MDHAR* regulates the expression of *GalLDH*, *GalUR*, and *DHAR* genes, positive regulatory effects of these genes on AsA synthesis have been reported in various species, including *Arabidopsis* [39], strawberry [26], and kiwifruit [17]. Therefore, it is suggested that *FaMDHAR50* may promote the accumulation of AsA by upregulating the expression of *FaGalLDH*, *FaGalUR*, and *FaDHAR*.

In addition to its positive role in promoting the accumulation of AsA in plants, *MDHAR* was also found to play a role in regulating strawberry fruit ripening. Strawberry is a typical non-climacteric fruit, and its ripening is regulated by ABA and sucrose [40]. *NCED1* is a key regulatory gene in the ABA pathway, and silencing *FaNCED1* can lead to a decrease in ABA levels, which in turn can delay the ripening of strawberry fruit [41]. In this experiment, the expression level of *FaNCED1* was significantly increased. Additionally, overexpressing *FaMDHAR50* decreased the firmness of strawberries and increased the expression level of *FaCEL1* compared to the control (Figure 7F), indicating that *FaMDHAR50* may enhance cell wall degradation and fruit softening by promoting the expression of *FaCEL1*. Meanwhile, sucrose is one of the key signaling molecules for strawberry fruit ripening, and it is the main sugar that accumulates in ripe strawberry fruit [41]. In this study, we also found that overexpression of *FaMDHAR50* promoted the accumulation of sucrose, glucose, and fructose in strawberry fruit (Figure 6G), and the expression of *FaSNS* and *FaSPS* was significantly increased. In *Solanum lycopodium* L., *MDHAR* has also been found to play a role in sugar metabolism [42]. As the fruit ripens, the starch present in the fruit gradually hydrolyzes into glucose [43], which serves as the initial reactant for the L-galactose pathway to synthesize AsA. Therefore, the increase in glucose accumulation in *FaMDHAR50*-overexpressed fruit may potentially contribute to the enhancement of AsA synthesis. Furthermore, it is known that organic acid accumulation in strawberry fruit is mainly dominated by citric acid [44], which decreases as the fruit ripens [45]. Overexpression of *FaMDHAR50* in strawberry fruit significantly reduced the content of citric acid and the expression levels of *FaCS* and *FaACL* (Figure 7I,J), indicating that *FaMDHAR50* may regulate the expression levels of *FaCS* and *FaACL* to decrease citric acid content in strawberry fruit. Taken together, these findings demonstrated that *FaMDHAR50* contributes to the degradation of the cell wall and citric acid, as well as the sugar accumulation during strawberry fruit ripening. 

Anthocyanins are the most abundant flavonoid compounds in strawberry fruit. The main types of anthocyanins in strawberry fruit are pelargonidin and cyanidin, with pelargonidin 3-glucoside being the predominant one [46]. In our study, we observed that overexpressing *FaMDHAR50* in strawberry fruits significantly decreased the content of pelargonidin 3-glucoside, while increasing the content of cyanidin chloride. The total anthocyanin content was reduced by 10% compared to the control (Figure 6F). These results are consistent with the inhibited pigmentation on the fruit surface (Figure 6A), as well as the significantly decreased *a** value (Figure 6A,D). In the flavonoid metabolic pathway, dihydrokaempferol can be catalyzed by F3′5′H and F3′H to generate the precursors of delphinidin and cyanidin, which are dihydromyricetin and dihydroquercetin, respectively. In this study, the increased expression of the *F3′5′H* gene may help to convert more dihydrokaempferol into the accumulation of delphinidin, which could be the main reason for the decrease in pelargonidin 3-glucoside content. Additionally, the flavonoid glycosyltransferase can catalyze the formation of stable anthocyanins from cyanidin. Therefore, the increase in UFGT expression also contributes to the accumulation of cyanidin chloride. Taken together, the role of *FaMDHAR50* in regulating the accumulation of different types of anthocyanins during strawberry fruit ripening varies. This also indicates that the role of *FaMDHAR50* in regulating the formation of fruit quality is not completely consistent with the typical changes that occur during fruit ripening, such as the accumulation of sugar and anthocyanin, softening of texture, and reduction in acidity levels. Instead, it involves sugar accumulation, texture softening, and decreases in anthocyanins and organic acid content, which implies better flavor when harvested before reaching full maturity. This is also in line with the quality standards for strawberry production, which usually require fruits to be harvested when they reach 70% to 90% ripeness.

## 4. Materials and Methods

### 4.1. Plant Materials

The strawberry cultivar ‘Benihoppe’ and its mutant material (Mutant, MT) were obtained from the strawberry cultivation base in Hanyuan County, Ya’an City, Sichuan Province, China. The fruit development stages were defined as small green (SG), large green (LG), white ripe (W), partial red (PR), and full red (FR) according to the number of days after flowering and fruit color. Fifteen strawberry fruits with uniform size, free of pests and diseases and mechanical damage, were collected separately at each developmental stage of LG, PR, and FR. Then, grind them into powder in liquid nitrogen and store them in an ultra-low temperature refrigerator at −80 °C for subsequent experiments. This experiment was repeated three times.

### 4.2. Physicochemical Characterization, Chromosomal Localization, Phylogenetic Analysis, and Promoter Analysis of the FaMDHAR Gene Family

To identify candidate *FaMDHAR* genes in the strawberry genome, a Hidden Markov Model of the *MDHAR* structural domain (PF00085) was used. A reference is made to Liu [47], who published the strawberry genome *Fragaria* × *ananassa* Camarosa Genome v1.0.a2 (Re-annotation of v1.0.a1). Searches were performed using the software HMM search default parameters, using CDD (http//www.ncbi.nlm.nih.gov/cdd/; accessed on 10 September 2022) and the SMART database (http://smart.embl-heidelberjg.de/; accessed on 10 September 2022) to verify the presence of MDHAR in all candidate MDHAR protein sequence structural domains. MEGA was used to construct a phylogenetic tree based on the neighbor joining (N–J) method with a bootstrap value of 1000. The tree was beautified by using the Evolview website. (https://evolgenius.info//evolview-v2/#login; accessed on 15 September 2022). Amino acid length, protein molecular weight, theoretical isoelectric point (pI), instability index, and total average hydrophilicity value (GRAVY) were predicted using the ProtParam tool (http://www.expasy.org/tools/protparam.html; accessed on 15 September 2022), along with the WoLF PSORT website (https://www.genscript.com/wolf-psort.html; accessed on 15 September 2022) for subcellular localization prediction analysis. Tbtools was used to analyze and visualize the promoter cis-acting elements of the *FaMDHAR* gene family.

### 4.3. RNA Extraction, cDNA Synthesis, and PCR Amplification

The total RNA was extracted following the CTAB (Cetyltrimethylammonium Bromide) method as described by Chen et al. [48], with slight modifications. First-strand cDNA was synthesized using Revert Aid H Minus reverse transcriptase (Thermo Fisher, Waltham, MA, USA) and random primers. All PCR amplifications were performed on a PT100 thermal cycler (Bio-Rad, Hercules, CA, USA). Twenty microliters of reactions were set up by combining 1 μL cDNA template, 1 μL forward and reverse gene-specific primers (Appendix A), and 10 μL PrimeSTAR Max premix (Takara Bio, Dalian, China). A total of 35 PCR reaction cycles were used, including one step at 98 °C for 10 s, 60 °C for 15 s, 72 °C for 2 min, and a final extension step at 72 °C for 5 min. The PCR amplification products were detected by electrophoresis in a 1% agarose gel.

### 4.4. Ribonucleic Acid Sequencing and Transcriptome Analysis

Eighteen libraries were constructed and sequenced by Novogene (Beijing, China). Briefly, mRNA was first enriched using oligo(dT) beads, followed by fragmentation in NEB fragmentation buffer. First- and second-strand cDNAs were synthesized in the presence of random hexamer primers, DNA polymerase I, and RNase H. Thereafter, a poly(A) tail and NEBNext adaptors were added to the two-stranded cDNAs. Next, cDNAs of size 250–300 base pairs (bp) were selected using 3 µL of USER Enzyme (NEB, Ipswich, MA, USA), adaptor-ligated cDNA at 37 °C for 15 min, followed by 5 min at 95 °C. Next, PCR amplifications were carried out using Phusion high-fidelity DNA polymerase and universal PCR primers. The quality of the libraries was assessed on an Agilent 2100 bioanalyzer. A total of 18 (three replicates for each stage of MT and WT fruits) libraries were clustered and sequenced (150 bp, pair-end) on a Hiseq-2500 platform. Low-quality (Q < 20) raw reads were screened by FASTQ software, and the adaptors were trimmed by using trim-galore (v0.6.6). Cleaned reads were then mapped onto the reference strawberry genome (v1.0.a2) and quantified using the Hisat 2 and string tie (v) pipelines with the default parameters [49]. To detect differentially expressed genes (DEGs), the DESeq 2 (v3.34.1) R package was utilized. Significant DEGs were defined as those with log2 transformed FC > 1 or <−1, and the adjusted *p* ≤ 0.05. The data presented in the study have been deposited in the NCBI SRA database under accession number PRJNA838938.

### 4.5. Transient Overexpression of FaMDHAR50 in Strawberry Fruits

The full-length coding sequence (CDS) of *FaMDHAR50* was amplified with primer pairs oe*FaMDHAR50*-F and oe*FaMDHAR50*-R (Appendix A). The amplicon was then inserted downstream of the 35S promoter in the modified binary vector *pCambia1301*. The sequencing-confirmed constructs were introduced into strawberry fruit at the degreening stage (19DPA) by *Agrobacterium*-mediated genetic transformation according to the method of Lin et al. [50]. The surface of the strawberry fruit was first disinfected with 75% ethanol for 5 s, followed by 0.2% AMISTAR 20 (Syngenta) for 2 s. The cut end of the stem was wrapped in disinfected wet cotton, and the *Agrobacterium* suspension was injected slowly and uniformly into the fruit until it was fully infected. The infiltrated fruits were incubated in a growth chamber at 23 °C, white LED light (16 h per day), and a relative humidity of approximately 90%. Five days after infiltration, fresh samples were collected before fruiting for hardness and color difference measurements, and subsequently, the fruit was cut into blocks, mixed, and stored in liquid nitrogen at −80 °C refrigerator until further use. There were ten fruits per replicate, and the experiment was repeated three times.

### 4.6. Determination of Color Difference and Hardness of Strawberry Fruit

Fruit color was measured using a colorimeter (CR-400, Konica Minolta, Tokyo, Japan). The results were expressed as *a** value. Fruit hardness was measured using a hardness tester (FR-5105, LUTRON, Guangzhou, China).

### 4.7. Determination of Ascorbic Acid and Dehydroascorbic Acid Content in Strawberry Fruits

The determination of ascorbic acid and dehydroascorbic acid content was referred to the method of Miyazawa et al. [51]. The supernatant was extracted by ultrasonication for 30 min with 10 mL of metaphosphoric acid solution (20 g/L) and centrifuged at 10,000 r/min. Five milliliters of the supernatant was filtered through a 0.45 nm aqueous filter tip into the injection bottle for the determination of L-ascorbic acid content; another 1 mL of the extract was added with 500 uL of cysteine (80 g/L) and stirred thoroughly for 2 min. The pH was adjusted to 7.0 with sodium phosphate (200 g/L), then adjusted to 2.5–2.8 with metaphosphoric acid, and finally the volume with was fixed water to 20 mL, then filtered with a 0.45 nm aqueous filter tip. The L-ascorbic acid content is then determined. The total ascorbic acid content of the reduced dehydroascorbic acid and the original L-ascorbic acid is obtained by subtracting the L-ascorbic acid content measured for the first time. The content of dehydroascorbic acid can then be determined. The standard working solution of L-ascorbic acid was prepared by dissolving the standard in metaphosphoric acid (20 g/L) in varying concentrations of 100 mg/L, 50 mg/L, 25 mg/L, 12.5 mg/L, 6.25 mg/L, and 3.125 mg/L, which was then ready to use. The determination conditions were as follows: the column used was a c column (4.6 mm × 250 mm, 5 m); the detection wavelength was set at 245 nm; the column temperature was 30 °C; the mobile phase consisted of methanol-phosphate-buffered solution (0.01 mol/L dipotassium hydrogen phosphate solution, adjusted to pH 3.5 with phosphoric acid) in a volume ratio of 2:98; the flow rate was set at 0.5 mL/min; and the injection volume was 10 uL. The HPLC standards were purchased from Sigma (Cibolo, TX, USA).

### 4.8. Determination of Anthocyanin Content in Strawberry Fruits

Anthocyanin content in strawberry fruits was determined following the method of Donno et al. [52], which involved using 1% HCL methanol solution and 30% methanol to extract anthocyanins, respectively. The anthocyanins were subsequently detected at a wavelength of 510 using an Agilent HPLC system with a DAD detector and a Silgreen ODS C18 column. The detection of anthocyanidins was achieved using a linear gradient elution procedure, where a 5% formic acid aqueous solution was used as eluent A, and methanol was used as eluent B. The procedure involved 100–0% A/B for 20 min, followed by 100% B for 5 min. Isocratic elution was performed using acetonitrile/water 12:88 (*v*/*v*) and 0.1% formic acid as the mobile phase for 16 min, and the sample was injected at 10 µL, with a flow rate of 1 mL/min. The HPLC grade standards were purchased from Sigma (USA).

### 4.9. Determination of Sugar Content of Strawberry Fruits

Glucose, fructose, and sucrose were determined using the method described by Xu et al. [53]. A 0.5 g sample was homogenized with 4.0 mL ultrapure water and incubated at 80 °C for 15 min. The supernatant was collected and centrifuged at 8000× *g* for 5 min at room temperature. The resulting solution was filtered through a 0.45 um cellulose acetate filter using an Agilent 1260 instrument equipped with a refractive index detector and a CNW NH2-RP column (4.6 mm × 250 mm, 5 μm, Anpu Technology Co., Ltd., Shanghai, China). The mobile phase consisted of a mixture of acetonitrile and water (80:20, *v*/*v*), with a flow rate of 1 mL/min. The column temperature was 25 °C, while the detector temperature was set at 40 °C. An injection volume of 10 uL was used for the analysis. HPLC-grade standard products were purchased from Sigma (USA).

### 4.10. Determination of Citric Acid Content in Strawberry Fruits

A 0.5 g portion of the sample was first extracted with 4 mL of 0.2% phosphate water and subsequently detected on an Athena C18-WP column using an Agilent HPLC system (eluent procedure: 3% methanol and 97% phosphate water (0.2%), 10 min). The content of citric acids was quantified by comparison with the corresponding external standards. The experiment was repeated three times independently. All HPLC grade standards were purchased from Sigma (USA).

### 4.11. Real-Time Fluorescence Quantification

The real-time quantitative PCR (qPCR) technique was used to detect the expression levels of the relevant genes in strawberry fruits (Appendix A). A 10 µL reaction system was established, including a 1 µL cDNA template, a 1 µL gene-specific primer pair (Appendix A), and a 5 µL TB green premix Ex Taq II (TaKaRa, Dalian, China). A three-step PCR reaction was performed, in which the denaturation step was performed at 94 °C for 30 s, the annealing step at 58 °C for 10 s, and the extension step at 72 °C for 10 s. The total reaction cycle was set to 40. The FaActin2 gene (LOC101313255) was selected as an internal control. The relative expression levels of the detected genes were calculated by the 2−ΔΔCt method. Three wells of each sample were used as three technical replicates, and all qPCR reactions were performed with three independent biological replicates.

### 4.12. Statistical Analysis of Data

IBM SPSS statistical software (v25.0) was used for the statistical analysis of the data, and the LSD method was used for significance analysis. The data were expressed as the mean ± standard deviation. The experiment was repeated three times.

## 5. Conclusions

Overall, we identified 168 FaMDHAR genes in the octoploid strawberry genome and discovered a crucial gene, *FaMDHAR50*, that positively regulates the accumulation of ascorbic acid in strawberry fruit. This gene not only promotes ascorbic acid accumulation by increasing the expression of *FaGalLDH*, *FaGalUR*, and *FaDHAR*, but also regulates fruit ripening quality by promoting fruit softening, increasing sugar accumulation, and decreasing citric acid and anthocyanin content. The identification and functional analysis of this gene provide a potential genetic resource for breeding high-Vc strawberry cultivars, and our findings shed new light on the involvement of the key gene *FaMDHAR50* in the formation of strawberry fruit quality.

## Figures and Tables

**Figure 1 ijms-24-09510-f001:**
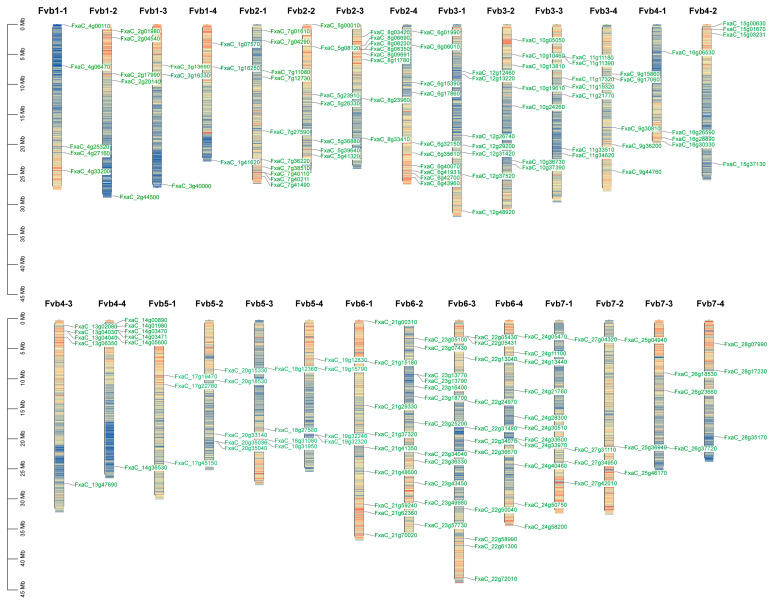
Localization and distribution of the MDHAR gene in strawberries on chromosomes.

**Figure 2 ijms-24-09510-f002:**
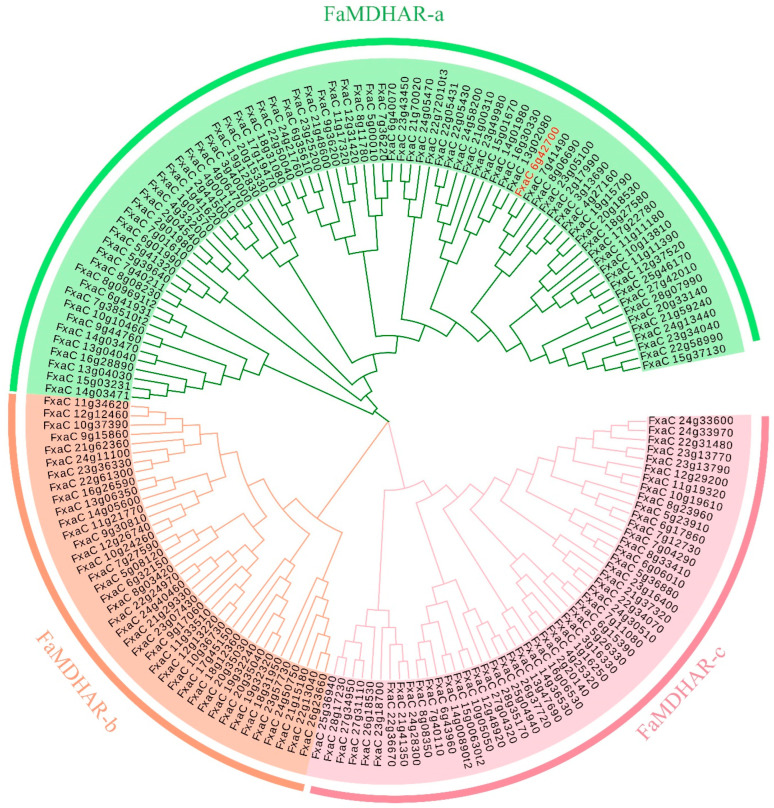
Phylogenetic analysis of the MDHAR gene in strawberries. Green fan: MDHAR-a; orange fan: MDHAR-b; pink fan: MDHAR-c. The red gene is a function verification gene for subsequent transient injection.

**Figure 3 ijms-24-09510-f003:**
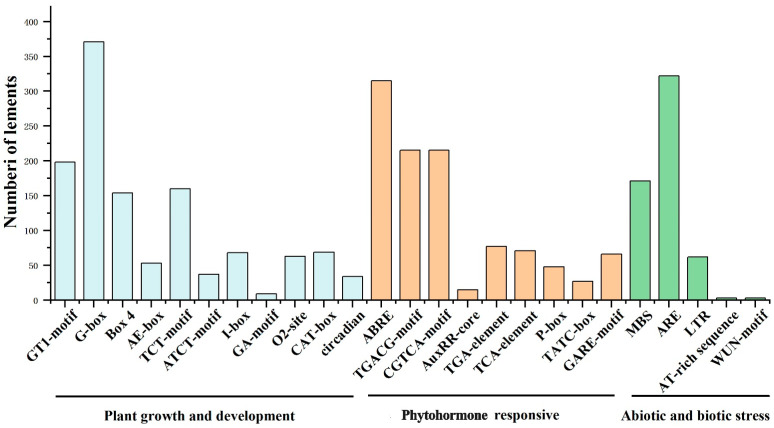
Cis-acting elements in the promoter regions of *FaMDHAR* genes. Statistical histogram of the number of plant growth and development, plant hormone response, and biotic and abiotic stress response elements.

**Figure 4 ijms-24-09510-f004:**
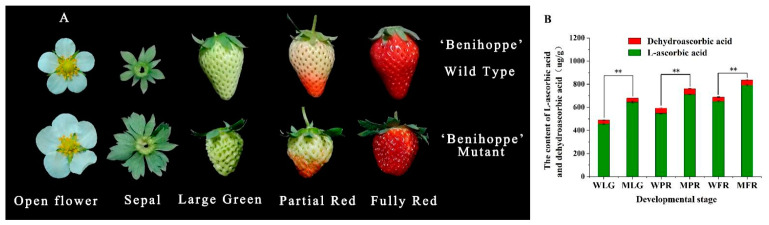
The differences between ‘Benihoppe’ and its mutant in phenotype and ascorbic acid content. (**A**) The phenotypic differences in flowers, sepals and fruits. (**B**) The differences in ascorbic acid and dehydroascorbic acid content. WLG, WT large green; WPR WT partial red; WFR fully red. MLG, MT large green; MPR, MT partial red; MFR, MT fully red. This value is expressed as the average ± SD of three biological replicates and three technical replicates. ‘**’ represents a highly significant difference between the total ascorbic acid content at *p* ≤ 0.01.

**Figure 5 ijms-24-09510-f005:**
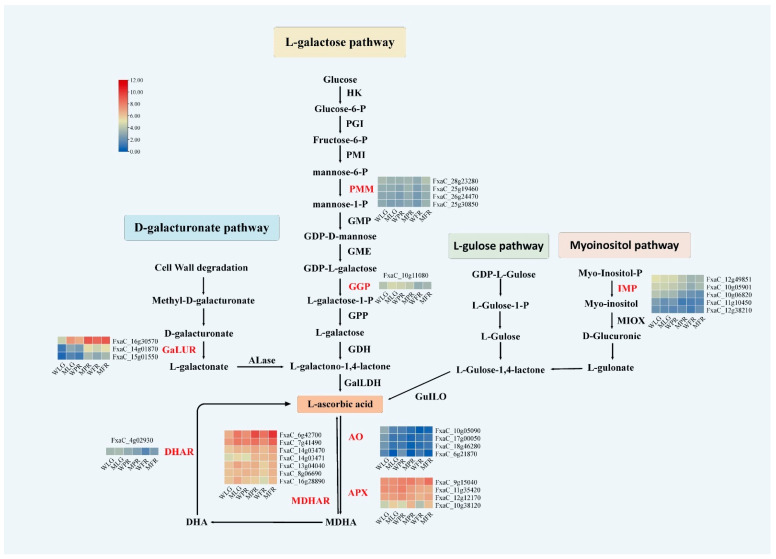
Transcriptional expression analysis of the AsA metabolic pathway in strawberries. The main pathways for ascorbic acid synthesis in plants are the L-galactose pathway, the myoinositol pathway, the L-gulose pathway, and the D-galacturonate pathway. *Alase*, aldono-lactonase; *AO*, ascorbate oxidase; *APX*, ascorbate peroxidase; DHA, dehydroascorbic acid; *DHAR*, dehydroascorbate reductase; *GalUR*, D-galacturonate reductase; *GDH*, L-galactose dehydrogenase; *GGP*, GDP-L-galactose-phosphorylase; *GalLDH*, L-galactono-1,4-lactone dehydrogenase; *GMP*, GDP-mannose pyrophosphorylase; *GPP*, L-galactose-1-phosphate phosphatase; *GulLO,* L-gulono-1,4-lactone oxidase; *HK*, hexokinase; *IMP*, myoinositol monophosphatase; MDHA, monodehydroascorbic acid; MDHAR, monodehydroascorbate reductase; *MIOX*, myoinositol oxygenase; *PGI*, phosphoglucose isomerase; *PMI*, phosphomannose isomerase; *PMM*, phosphomannomutase. For the gene expression comparison, the log2 transformed CPM values were presented as a heatmap with the WT on the left and the OE of MT on the right.

**Figure 6 ijms-24-09510-f006:**
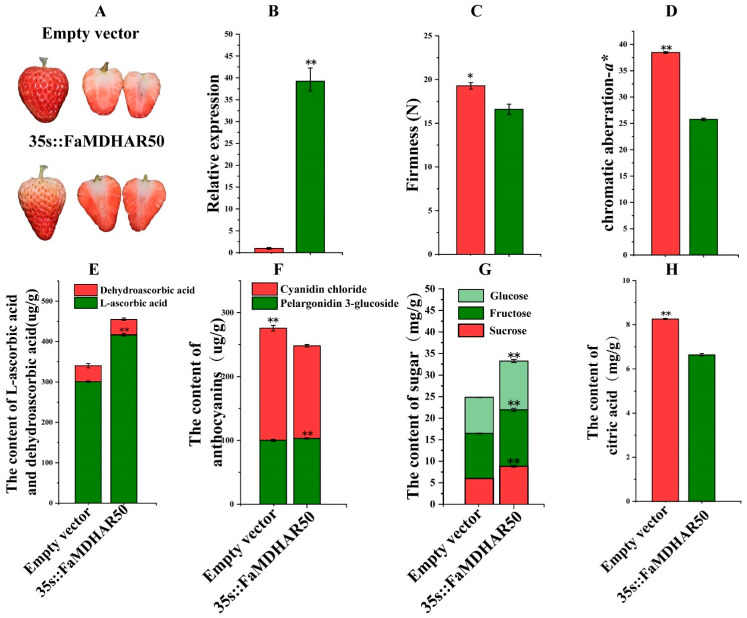
Transient overexpression of *FaMDHAR50* in strawberry fruits. (**A**) The appearance of the treated fruits five days after Agrobacterium infiltration; (**B**) the relative expression of FaMDHAR50 and in the fruit at 5 dpi; (**C**) fruit firmness; (**D**) chromatic aberration-*a**; (**E**) the content of L-ascorbic acid and dehydroascorbic acid; (**F**) the anthocyanin content; (**G**) the amount of sugar; (**H**) the content of citric acid. The values are presented as the mean ± SD of three biological and three technical replicates. ‘*’ represents a highly significant difference at *p* ≤ 0.05, ‘**’ represents a highly significant difference at *p* ≤ 0.01.

**Figure 7 ijms-24-09510-f007:**
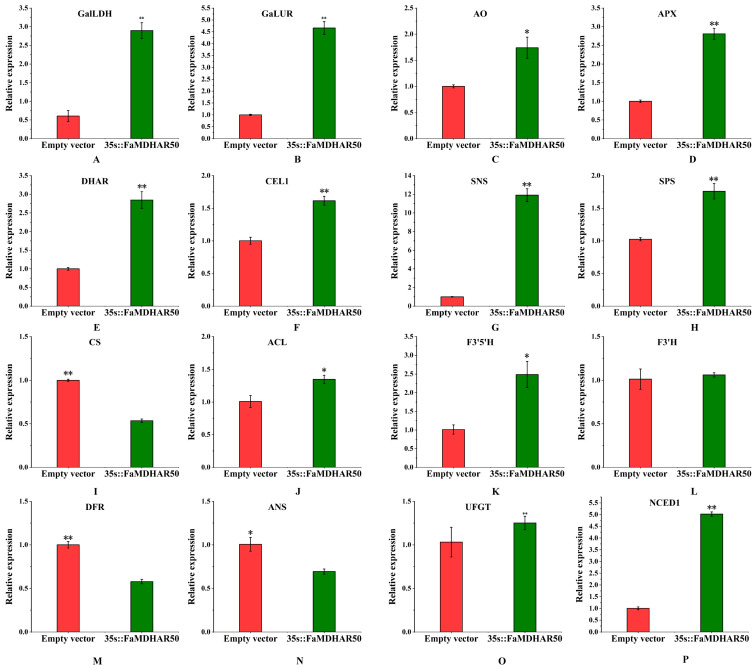
The expression of some related genes after overexpression of *FaMDHAR50*. (**A**–**P**): *GalLDH*, L-galactono-1,4-lactone dehydrogenase; *GalUR*, D-galacturonate reductase; *AO*, ascorbate oxidase; *APX*, ascorbate peroxidase; *DHAR*, dehydroascorbate reductase; *CEL1*, endo-1,4-beta-glucanase; *SNS*, sucrose synthase; *SPS*, sucrose phosphate synthase; *CS*, citrate synthetase; *ACL*, citrate lyase; *F3′5′H*, flavonoid 3′,5′-hydroxylase; *F3′H*, flavonoid 3′-hydroxylase; DFR, dihydroflavonol 4-reductase; *ANS*, anthocyanidin synthase; *UFGT*, UDP-glycose flavonoid glycosyltransferase; *GPI*, 6-glucose phosphate isomerase. *NCED1*, 9-cis-epoxycarotenoid dioxygenase. 1; ‘*’ represents a highly significant difference at *p* ≤ 0.05, ‘**’ represents a highly significant difference at *p* ≤ 0.01.

## Data Availability

The genome sequence data is available from the GDR database (accessed on 10 September 2022). The data presented in the study have been deposited in the NCBI SRA database under accession number PRJNA838938.

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
