# Peer review of "Genome-Wide Identification and Comparative Transcriptome Methods Reveal FaMDHAR50 Regulating Ascorbic Acid Regeneration and Quality Formation of Strawberry Fruits"

_ijms, 2023, doi:10.3390/ijms24119510_

Round 1

Reviewer 1 Report

Review

Ascorbic acid (AsA) is a powerful antioxidant known to play a vital role in both plant and human life. Therefore, the study by Hou and colleagues on the analysis of the key genes of AsA biosynthesis in the "queen of fruit" strawberriy is of significant practical importance. Using genome-wide identification and a comparative transcriptome approach the authors showed that higher AsA content and quality formation of strawberry fruit in the natural mutant of ‘Benihoppe’ cultivar was  due to  FaMDHAR50  encoding monodehydroascorbate reductase, the enzyme of AsA-GSH cycle, a key  pathway involved  in the regeneration of oxidized AsA. To explore the potential biological functions of the FaMDHAR gene, the authors undertook an analysis of cis-acting elements in the promoter region which revealed cis-elements implicated in biotic  and abiotic stress responses, as well as in the defense and repair processes associated with  trauma and pathogen invasion. Transcriptome analysis was confirmed by transient overexpression of FaMDHAR50 combined with quantification of fruit quality-related genes by Real Time PCR.

The relevance of the study is beyond doubt. The manuscript is logically structured and corresponds to the results obtained  

I have only several minor concerns:

1.     Lines  17,176-177, 305  

It should be clarified that the authors mean the products of the genes, and not the genes themselves, since the genes in question are localized in the genome, and not in the plastome, chondriome, or vesicles.

2.     Line 155 Please, change Phytohormonr  to Phytohormone

3.     Lines 177,255,293. Please, add “difference” after “significant”

Pleate, use italics throughout when referring to a gene and not uts product

Editing by a native English speaker is always welcome

Author Response

Response to Reviewer 3 Comments

Point 1: It should be clarified that the authors mean the products of the genes, and not the genes themselves, since the genes in question are localized in the genome, and not in the plastome, chondriome, or vesicles.

Response 1: Thank you for your valuable suggestion. We accepted your opinion and made it clear that it is indeed the products of genes that are located in various organelle.

Point 2: Line 155 Please, change Phytohormonr  to Phytohormone

Response 2: Special thanks to you for your good comments. We have made modifications in the manuscript.

要点3:177,255,293行。请在“显著”之后加上“差异”

响应 3:感谢您的宝贵反馈。我们对手稿进行了修改。

Reviewer 2 Report

The authors identified and explored the influence of MDHAR genes on the synthesis of an ascorbic acid. The increase of vitamin C by over expression of MDHAR gene is known on other species, so can be expected also on strawberries. Experimental verification of overexpression of a potential MDHAR gene is positively separates this work against many similar works.

Minor suggestions:  I would remove figure 2 and figure 3A with respective paragraphs, because it gives no information on functionality of predicted genes, and regulatory elements prediction, as it is done, makes no sense and there are no either experimental or any other proof of correctness of elements predictions. 

Overall the paper needs minor English corrections: Structural  tree  beautification; stored in liquid nitrogen at -80 °C.

Author Response

Response to Reviewer 1 Comments

Point 1: I would remove figure 2 and figure 3A with respective paragraphs, because it gives no information on functionality of predicted genes, and regulatory elements prediction, as it is done, makes no sense and there are no either experimental or any other proof of correctness of elements predictions. 

Response 1: Special thanks to you for your good comments. Your comments are all valuable and very helpful for revising and improving our paper, as well as the important guiding significance to our research. We have studied the comments carefully and have made corrections which we hope meet with approval. Revised portion are marked in red in the paper.

We believe that phylogenetic analysis of MDHAR genes (Figure. 2) is necessary. At present, there is no analysis of the whole gene family of MDHAR genes in strawberry fruit. Clarifying the distance and grouping of genetic relationships is beneficial for better screening and functional prediction of genes. We have also deleted Figure 3A and its related paragraphs, they do not have any experiments or other evidence to prove the correctness of component predictions, as stated in your opinion. Thank you again for your valuable feedback.

Reviewer 3 Report

I have only minor comments on this submission.

There is a confusion between gene and gene product in several places in the ms.

Line 19: “168 genes and most of them are predicted to exist in the chloroplast…”

It should be “gene product” or “encoded proteins”.

Same in line 120.

Section 2.3. it should be figure 3, not 4. Fig. 3B is not cited.

Section 2.4. it should be figure 4, not 5.

Section 2.5. No mention of Fig. 5.

Figure 5. It is unfortunate that the order of samples (WLG, WFR, etc.) is not the same as in fig. 4B (WLG, MLG, etc.). It makes it more difficult to compare the 2 lines. I suggest to change it in fig.5.

Discussion, lines 317-320. The recommended dietary allowance of VitA is not only relying on strawberries. Therefore even the lowest content of 50 mg/100g will make a valuable contribution. Therefore the sentence about “most strawberry varieties not meeting this standard of 80 mg/100g…” is a little bit strange.

Author Response

Response to Reviewer 4 Comments

Point 1: There is a confusion between gene and gene product in several places in the ms.

Line 19: “168 genes and most of them are predicted to exist in the chloroplast…”

It should be “gene product” or “encoded proteins”.

Same in line 120.

Response 1: Special thanks to you for your good comments. We do have the problem of confusing genes and gene products. We have made changes in the resubmitted manuscript, such as lines 19-21, 54-55, and 242. In line 120, Section 2.5, is the analysis of transcriptome data, so it is expressed as transcript in the text, which we think is more accurate. If you have any better feedback, we look forward to your reply. Thank you again for your valuable feedback.

Point 2: Section 2.3. it should be figure 3, not 4. Fig. 3B is not cited. Section 2.4. it should be figure 4, not 5. Section 2.5. No mention of Fig. 5.

Response 2: Thank you very much for discovering our negligence, which is very beneficial for us. We made changes to the resubmitted manuscript. Figure 5 has been added on line 118. Thank you again for your valuable feedback.

Point 3: Figure 5. It is unfortunate that the order of samples (WLG, WFR, etc.) is not the same as in fig. 4B (WLG, MLG, etc.). It makes it more difficult to compare the 2 lines. I suggest to change it in fig.5.

Response 3: Your opinion is very useful, as unifying the stage order of Figures 4 and 5 provides more convenience for readers when reading. We have made changes to the order of the heat maps in Figure 5 according to your suggestion. You can refer to it in the resubmitted manuscript. Thank you again for your valuable feedback.

Point 4: Discussion, lines 317-320. The recommended dietary allowance of VitA is not only relying on strawberries. Therefore even the lowest content of 50 mg/100g will make a valuable contribution. Therefore the sentence about “most strawberry varieties not meeting this standard of 80 mg/100g…” is a little bit strange.

Response 4: This is indeed our narrow understanding of the literature. Thank you for pointing out the correction errors. We have made new statements on lines 255-256 of the resubmitted manuscript. Thank you again for your valuable feedback.